# How to Distinguish Marfan Syndrome from Marfanoid Habitus in a Physical Examination—Comparison of External Features in Patients with Marfan Syndrome and Marfanoid Habitus

**DOI:** 10.3390/ijerph19020772

**Published:** 2022-01-11

**Authors:** Lidia Wozniak-Mielczarek, Michalina Osowicka, Alicja Radtke-Lysek, Magda Drezek-Nojowicz, Natasza Gilis-Malinowska, Anna Sabiniewicz, Maksymilian Mielczarek, Robert Sabiniewicz

**Affiliations:** 1Department of Pediatric Cardiology and Congenital Heart Defects, Medical University of Gdansk, 80-210 Gdansk, Poland; sabini@gumed.edu.pl; 2Department of History of Medicine, Medical University of Warsaw, 02-091 Warsaw, Poland; michalina.osowicka@wum.edu.pl; 31st Department of Cardiology, Medical University of Gdansk, 80-210 Gdansk, Poland; alaradtke@gumed.edu.pl (A.R.-L.); tasza.gilis@gmail.com (N.G.-M.); max.mielczarek@gmail.com (M.M.); 4Department of Ophthalmology, Medical University of Gdansk, 80-210 Gdansk, Poland; magda.drezek@op.pl; 5Students’ Scientific Circle of the Department of Pediatric Cardiology and Congenital Heart Diseases, Medical University of Gdansk, 80-210 Gdansk, Poland; a.sabiniewicz@gumed.edu.pl

**Keywords:** Marfan syndrome, marfanoid habitus, thumb sign, wrist sign, pectus carinatum, pectus excavatum, gothic palate, Ghent criteria, aortic root dilatation

## Abstract

Marfan Syndrome (MFS) is a systemic disorder caused by mutations in fibrillin-1. The most common cause of mortality in MFS is dissection and rupture of the aorta. Due to a highly variable and age-dependent clinical spectrum, the diagnosis of MFS still remains sophisticated. The aim of the study was to determine if there exist phenotypic features that can play the role of “red flags” in cases of MFS suspicion. The study population included 306 patients (199 children and 107 adults) who were referred to the Department of Pediatric Cardiology due to suspicion of MFS. All patients underwent complete clinical evaluation in order to confirm the diagnosis of MFS according to the modified Ghent criteria. MFS was diagnosed in 109 patients and marfanoid habitus in 168 patients. The study excluded 29 patients with other hereditary thoracic aneurysm syndromes. Comparative analysis between patients with Marfan syndrome and marfanoid habitus was performed. Symptoms with high prevalence and high positive likelihood ratio were identified (pectus carinatum, reduced elbow extension, hindfoot deformity, gothic palate, downslanting palpebral fissures, lens subluxation, myopia ≥ 3 dioptres remarkably high stature). The differentiation between patients with MFS and marfanoid body habitus is not possible by only assessing external body features; however, “red flags” could be helpful in the screening phase.

## 1. Introduction

Marfan Syndrome (MFS) is an autosomal dominant systemic disorder caused by mutations in the extracellular matrix protein: fibrillin-1 (FBN1). The estimated incidence of MFS is 2–3 per 10,000 individuals [1]. MFS affects the connective tissue of multiple organs and systems such as the adipose and muscle tissues, skin, pulmonary and central nervous system, but the three most susceptible systems are the: cardiovascular, ocular and skeletal systems [1,2]. The clinical spectrum of MFS is highly variable, from a mild form (involving one or a few systems) to a severe and rapidly progressive neonatal type. The most common cause of mortality in MFS is progressive aortic root dilatation, which can lead to dissection and rupture of the aorta [3]. Early diagnosis and specialized care, which includes limitation of physical activities, pharmacotherapy using β-blockers or antagonists of angiotensin receptors (ARB), may delay the evolution of aortic aneurysms [4,5]. Moreover, regular echocardiographic follow-ups allow for preventive aortic root surgery before a likely life-threatening event.

Diagnosing MFS is difficult and requires multidisciplinary assessment of well-experienced physicians, such as cardiologists, ophthalmologists, orthopedists and geneticists. Molecular testing for FBN1 mutations is also available, but an unequivocal diagnosis of MFS requires more than just the demonstration of gene mutation. The clinical criteria used for MFS diagnosis—the revised Ghent nosology—does not only include the identification of FBN1 gene mutation but also takes into consideration positive family history, aortic root dilatation and lens dislocation. In inconclusive situations, e.g., dilatation of the aortic root in the absence of dislocation of the lens and negative family history or positive family history without dilatation of the aortic root and dislocation of the lens, additional features of MFS found in the systemic score are taken into account [6]. The most common symptoms that lead to suspicion of MFS are external features and deviations noted during physical examination such as remarkably high stature, asthenic body structure, dolichostenomelia, arachnodactyly, chest deformities, characteristic facial attributes and other abnormalities. Unfortunately, often, some of these features are still overlooked by primary care physicians, pediatricians, internists, orthopedists and other caregivers. This has consequently resulted in patients not being referred for MFS diagnosis, which can then result in the endangerment of their lives. On the other hand, there are also patients (referred to as marfanoid types) who exhibit some of the body features that are characteristic for MFS (e.g., tall and slim figure) who are unnecessarily recommended for diagnosis. These patients and their families are therefore exposed to stress and a series of tests that may lead to stigmatization, life limitations and psychological burdens [7].

The aim of the study was to evaluate which external features should alert to the possibility of MFS in patients and if it is possible to distinguish MFS patients from marfanoid patients by only using a physical examination.

## 2. Materials and Methods

From January 2015 to January 2021, 306 patients (199 children and 107 adults) aged 2 months to 65 years were referred to the Department of Pediatric Cardiology and Congenital Heart Diseases due to suspicion of MFS. The most common reasons for suspecting MFS were: asthenic physique (tall and slim patients) (28%), family history of MFS (21%), joint laxity (16%), chest deformity (16%), lens subluxation (6%), scoliosis (5%), and dilation of the ascending aorta (5%) (Figure 1).

Patients were most often referred for diagnosis by doctors of the following specializations: cardiologists (22%), orthopedists (21%), pediatricians (11%), geneticists (7%) and family physicians (4%). It is also important to point out that in as many as 19% of patients, it was the parents who had such a suspicion (in 11% of patients due to family burden and in 8% for reasons other than family history) (Figure 2).

All patients were subjected to a complete clinical assessment, which included a detailed medical history encompassing family medical history, physical examination covering anthropometric measurements and cardiac examination (electrocardiography, 24 h ambulatory electrocardiographic monitoring and transthoracic echocardiography) as well as ophthalmologic, orthopedic and genetic consultations. Lastly, the modified Ghent criteria were then used to identify patients with MFS [6]. In those patients where the diagnosis was inconclusive, genetic tests were performed using the NGS method. The study was approved by the local Ethics Committee.

### 2.1. Detailed Description of the Medical History Taking Process and Physical Examination

The medical history included, among other data: the course of the prenatal period, neonatal period (including postnatal anthropometric measurements), infancy period, taking into account motor development, childhood development, chronic diseases, surgical procedures, medications taken, specialist care, the occurrence of pleural edema, hernias, the incidence of injuries (bone fractures, joint dislocations), reported symptoms (e.g., fainting, chest pain, palpitations, or irregular heartbeat, joint pain, coordination disorders, frequent headaches or dizziness), effort tolerance and detailed family history (regarding, among other things, the occurrence of all irregularities related to Marfan syndrome; in the absence of complete data, parents were also subjected to a comprehensive assessment in this regard).

Apart from the classic pediatric and internal medicine evaluations, the physical examination also included the assessment of body weight, height, length of the lower body segment and arm span, followed by the calculation of body mass index (BMI), body surface area (BSA), upper segment to lower segment ratio (US/LS ratio) and arm span to height ratio (ASHR). Furthermore, patients were examined for the presence of arachnodactyly (thumb and wrist sign), joint laxity, asymmetry and deformation of the chest (pectus carinatum or pectus excavatum) and scoliosis, including the assessment of the degree of severity and kyphosis. In addition, joint hyperextension, reduced elbow extension, dolichocephaly, features of facial dysmorphia (malar hypoplasia, enophthalmia, downslanting palpebral fissures, gothic palate, micrognation, retrognathia, dental crowding), stretch marks and flat feet and hindfoot deformity (deformation of the tarsus/medial displacement of the medial ankle) were also assessed. For differential diagnosis purposes, features characteristic of other marfanoid syndromes, such as Loeys–Dietz or Ehlers–Danlos syndrome, were also assessed. These included the degree of hypermobility according to the Beighton scale, Cleft palate or uvula, presence of craniosynostosis, hypertelorism, hyperelastic skin, low-set ears, auricular malformations and thin and translucent skin. The description of the methods of evaluation of the selected parameters from the physical examination is presented in Table 1.

### 2.2. Statistical Analysis

Continuous data were presented as a mean value and standard deviation (SD), while categorical data were presented as percentages. Normal distribution was verified by Kolmogorov–Smirnov test. Continuous data were compared by Student’s t-test or by Mann–Whitney U test, depending on their distribution. Categorical data were compared by using the Chi-square test or Fisher’s exact test. In order to find an independent risk factor for Marfan Syndrome diagnosis, we performed a univariate logistic regression, after which variables with *p* < 0.10 were analyzed using multivariate logistic regression. A *p* value less than 0.05 was considered statistically significant. Data were analyzed using SPSS software v.21 (IBM, Chicago, IL, USA).

## 3. Results

### 3.1. Patient Characteristics

Of the 306 patients referred with suspected MFS, 277 patients were incorporated into the final analysis: 109 were diagnosed with MFS (in accordance with the Ghent criteria), and 168 were diagnosed with marfanoid habitus (presented only with some external features of Marfan syndrome). The study excluded 29 patients diagnosed with neonatal Marfan syndrome (*n* = 2), Ehlers–Danlos syndrome (*n* = 16), Loeys–Dietz syndrome (*n* = 7), ectopia lentis syndrome (*n* = 2) and MASS phenotype (*n* = 2)—a familial connective tissue disorder similar to MFS. (MASS is an acronym for features of the disorder that may be present: M—mitral valve prolapse, A—aortic root dilation, S—skin striae, S—skeletal features.) Among 306 patients we have examined, genetic testing was performed in 35: in 11, the diagnosis of Marfan syndrome was confirmed; in 7, the diagnosis of Loeys–Dietz syndrome was confirmed; in 5, the diagnosis of Ehlers–Danlos was confirmed; and in 12 patients, no pathogenic or potentially pathogenic mutation was found (these patients were diagnosed for mutations in the FBN1 gene). In the group of patients diagnosed with MFS, 46 were children and 63 adults. The mean age was 23.8 ± 15 years (from 3 to 63 years), and the percentage of women was 47.7%. In the group identified as having marfanoid habitus, 132 were children and 36 adults, the mean age was 16.9 ± 8.8 years (from 2 to 51 years) and the proportion of women was 45.8%.

### 3.2. Comparative Analysis

Firstly, a comparative analysis of the most frequently reported symptoms, physical examination abnormalities and other more frequently noted abnormalities between patients with MFS and marfanoid habitus was performed. For the purpose of a more detailed analysis, the above-mentioned features were compared in relation to the entire population, also taking into consideration the division of the population into children and adults (Table 2). In adults, the above-mentioned analysis was also performed separately in women and men subgroups (Appendix A).

Seven out of all the examined features turned out to be significantly more common in patients with MFS in all three populations: children, adults and in the combined population. These features are: pectus carinatum, reduced elbow extension, hindfoot deformity, gothic palate, downslanting palpebral fissures, lens subluxation and myopia ≥ 3 dioptres (Table 2). In addition, in the child population excessive growth, ASHR > 1.05, dental crowding and micrognation were also significantly more often in patients with MFS than in patients with marfanoid habitus only. In the adult group, joint laxity, thumb sign, stretch marks and enophthalmia were also notably more specific for MFS.

In the analysis of separated men and women subgroups, differentiating features were the same as in the general adult population, with one additional feature—remarkably high stature in the women subgroup (59.9% in the MFS group vs. 25% in the marfanoid group, *p* = 0.029).

Interestingly, several features, such as deficiency in weight, US/LS < 0.85, wrist sign, scoliosis, moderate or severe scoliosis, pectus excavatum, flat feet, dolichocephaly, retrognathia, malar hypoplasia, commonly regarded as being helpful in screening for MFS turned out to be of no use in differentiating MFS from marfanoid habitus. In the child population, there was also no difference in the incidence of features such as the thumb sign, joint laxity, stretch marks and enophthalmia between patients with MFS and marfanoid habitus. Moreover, in adults, features such as remarkably high stature, ASHR > 1.05, dental crowding and micrognathia did not have any significant role in the differentiation between MFS and marfanoid habitus.

Furthermore, there was also no noteworthy difference between the groups when considering characteristics such as birth weight, birth length, the incidence of idiopathic pulmonary edema, prevalence of multiple injuries (fractures or dislocations) and coordination disorders. Among the reported symptoms, dizziness, palpitations and lower effort tolerance than that of peers were significantly more common in MFS patients than in marfanoid—this, however, was not confirmed in a separate analysis of the child and adult population. Chest pain, syncopy, joint pain and frequent headaches were found to be of a similar occurrence in both groups.

### 3.3. Logistic Regression Study

The next step of analysis involved determining the predictive value of individual features in the diagnosis of MFS. For that purpose, the univariate logistic regression study was performed and included the following features: USLS < 0.85, ASHR > 1.05, thumb sign, wrist sign, pectus carinatum, pectus excavatum, scoliosis, joint laxity, reduced elbow extension, dolichocephaly, malar hypoplasia, enophthalmia, micrognation, retrognathia, downslanting palpebral fissures, gothic palate, dental crowding, stretch marks, hindfoot deformity, flat feet, myopia ≥ 3 dioptres, lens subluxation, excessive growth (remarkably high stature), deficiency in weight, hernias and idiopathic pulmonary edema (Appendix A).

Variables that met the *p* < 0.1 criterion in the univariate analysis were included in the multivariate analysis in order to establish independent predictors for the diagnosis of MFS (Table 3).

In the child population, the following features turned out to be independent predictors for MFS diagnosis in the multivariate analysis (*p* < 0.05): ASHR > 1.05 (Odds Ratio, OR = 11.48), hindfoot deformity (OR = 6.02), lens subluxation (OR = 78.91) and hernias (OR = 7.25). Respectively, in the adult group, these were: enophthalmia (OR= 9.6) and remarkably high stature (OR = 14.54), whereas, for the entire population (children and adults considered together), these were ASHR >1.05 (OR = 3.48), pectus carinatum (OR = 3), hindfoot deformity (OR = 4.54), lens subluxation (OR = 67.12), excessive growth (remarkably high stature) (OR = 5.46) and hernias (OR = 6.45) (Table 3).

### 3.4. Red Flags

Features that turned out to be useful for differentiating between MFS and marfanoid habitus in children and adults are summarized in Table 4. These features should be seen as “red flags” in the context of Marfan syndrome suspicion (Figure 3).

## 4. Discussion

Marfan syndrome is a disease with a severe clinical course. The most common cause of mortality in MFS is progressive aortic root dilatation, which can lead to dissection or rupture of the aorta. Aortic root dilatation is found in about 75% of children and 85% of adults with MFS [8]. The risk of rupture or dissection correlates with the increasing diameter of the aorta, especially after it exceeds 50 mm [3,9]. Early diagnosis and professional care, which include pharmacotherapy, limitation of physical activities, regular echocardiographic follow-ups and preventive aortic root surgery, reduce mortality [4]. Despite increasing knowledge about MFS, it still goes unrecognized in many people, and not infrequently, diagnosis is made at the time of aortic complications (dissection, ruptures) or after sudden death caused by the above-mentioned complications in family members. It is therefore very important to be vigilant and refer patients who show phenotypic features of MFS to cardiologists for a professional assessment. Once diagnosed with MFS, patients should be systematically controlled, and preventive actions need to be implemented, as adequate prophylaxis allows for the avoidance of the tragic consequences of the acute aortic syndrome [4]. This is why active searching for MFS features by physicians of variable specialties seems to be very crucial.

On the other hand, reckless suspicion of MFS may have serious consequences and impact patients’ lives. For children, hasty suspicion of MFS may mean exemption from physical education and restriction of physical activity with peers. For adults, it may lead to limitations at work, trouble with employment and reduction of daily activity. This can lead to exclusion from society and stigmatization. Secondly, the burden of a potentially mortal genetic disease leads to increased psychological aftermath, such as lower quality of life, frequent depressive episodes and anxiety compared to the general population [10,11]. Our experience shows that a lot of tall and thin adolescents are sent for cardiology consultation with suspicion of MFS, but their final diagnosis is negative. In our study group, out of 306 individuals referred with MFS suspicion, in 168 (54.9%) of them, the diagnosis was not confirmed. We believe that proper screening by pediatricians or general physicians would prevent most of these unnecessary referrals. This would be of benefit not only for the individuals, who would avoid stigmatization by classifying their body type as habitus marfanoid, but also the public health care system by unburdening it.

This is why the general aim of the study was to evaluate which external features should alert to the possibility of MFS in suspected individuals and if it is possible to distinguish MFS patients from marfanoid patients by only using a physical examination. In our study population, the most common reasons that lead to suspicion of MFS were external features and deviations noted during physical examination such as: asthenic physique (high stature and weight deficiency-28% of patients), joint laxity (16%), chest deformity (16%), lens subluxation (6%) and scoliosis (5%). The primary objective of the study was to determine if there are any single phenotypic features that could play the role of “red flags” in the event of MFS suspicion. We focused on physical features, which can be easily rated by physicians of all specialties. We found that neither of the phenotypic features is specific and sensitive enough for diagnosing MFS, but we identified the most characteristic, which could be called “red flags”. “Red flags” depend on the patient’s age—whether they are children or adults. Based on two statistical analyses (comparative analysis and multivariate regression analysis), altogether, 16 features were selected for the population of children and adults, which may constitute alarming signals for MFS suspicion and may be useful hints in differentiating between patients with marfanoid body habitus and MFS. Eight of these features were valid irrespective of a patients’ age, namely: pectus carinatum, reduced elbow extension, hindfoot deformity, gothic palate, downslanting palpebral fissures, lens subluxation, myopia ≥ 3 dioptres, remarkably high stature (excessive growth in children). In addition, for the child population, the following features were also included: ASHR > 1.05, dental crowding, micrognation and hernias, while for adults, these were: thumb sign, joint laxity, stretch marks and enophthalmia.

Interestingly, some signs commonly regarded as straightforward MFS signs, such as pectus excavatum, wrist sign, scoliosis, flat feet or deficiency in weight, were shown to occur with similar prevalence in MFS patients and patients with marfanoid body habitus. These findings may have significant clinical implications by facilitating the screening of patients with MFS suspicion, especially in the aspect of specificity. As a secondary objective, we have also shown that phenotypic changes can be only used as an entryway to a proper diagnosis. In order to ultimately diagnose MFS, a comprehensive assessment by a group of specialists needs to be arranged. Of course, proper diagnosis of MFS remains a challenge even for well-experienced physicians. Even with the application of the Ghent criteria, which seems to be a valuable tool, the whole process is complicated and requires the engagement of a multidisciplinary team composed of cardiologists, ophthalmologists, orthopedists and geneticists. Additionally, the profile of phenotypic features changes with age (which we documented in our study), making the diagnosis of MFS in some patients even more sophisticated. Thus, in some cases, expensive and not readily available tests, such as genetic analysis or magnetic resonance imaging (to assess the presence of dural ectasia), are needed.

To the best of our knowledge, this is the first study that has pointed out physical features linked to a high likelihood of MFS that can be readily assessed by physicians of all specialties. A prior study with a similar objective was performed by Mueller et al., in which symptoms of MFS with a high positive likelihood ratio and high prevalence were identified and stratified in a risk score named the “Kid-Short Marfan Score” (kid-SMS) [12]. However, most of the manifestations included in Kid-SMS needed echocardiography assessment by well-qualified pediatric cardiologists. This makes the scale available only for specialists. They also emphasized the importance of adequate diagnosis and avoidance of stigmatization and psychological burden due to chronic disease. In the Kid-SMS score, one of the high-risk manifestations was the combination of three skeletal features. In this study, it was demonstrated in 48% MFS patients and only in 3% non-MFS patients (*p* < 0.001). The positive likelihood ratio was 11.23 + 0.54. This shows that a combination of three skeletal features can be valuable for screening for MFS, as well as three phenotypic features, as we have shown in our study [12,13].

Another study, which focused on physical features, namely facial changes and their recognition by physicians, was conducted by Beverlie et al. In this study, the facial features found to be more prevalent in MFS than in the normal population were enophthalmos and downslanting palpebral as well as retrognathia, which is in accordance to our findings. They showed that facial features could be used as an initial screening tool but with low sensitivity. Thus, other diagnostic methods need to be completed [14].

Our finding that MFS features change with patient age was previously noted by others. Monteil et al. showed that, in the MFS group, Ghent criteria systemic scores increased by 0.32 ± 0.8 per year and in the non-MFS group decreased by 0.53 ± 0.79 per year [15]. At the final visit, the MFS group had significantly higher Ghent scores than the non-MFS group (*p* < 0.0001). Faivre et al. demonstrated in a group of 320 children with MFS that, predominantly, clinical manifestations of MFS increased with age (for example, pectus carinatum or excavatum, dolichostenomelia, scoliosis > 20°, pes planus or spondylolisthesis) [16]. Stheneur et al. described the evolution of the MFS phenotype with age and compared it with a population of children consulted due to an MFS suspicion. They revealed that among the clinical features in the MFS group, height > 3.3 SD carried the highest positive predictive value of 72% for MFS and a negative predictive value of 79%, while arm span/height ratio was higher in the MFS group (*p* < 0.0001) in all age strata. They also proved that the prevalence of some skeletal features, such as ASHR, pectus deformity, wrist sign and severity of scoliosis, increased with age in the MFS group. Prevalence of joint hypermobility and pes planus tended to decrease with age, whereas striae appeared toward the age of 10 years [17].

In a previous study that we conducted (Differences in Cardiovascular Manifestation of Marfan Syndrome Between Children and Adults), significant changes within the cardiovascular system in the MFS group related to the patients’ age were reported. In that study, in a group consisting of 101 patients with MFS (44 children, 57 adults), there was a significantly higher prevalence of aortic arch dilatation, descending thoracic and abdominal aorta dilatation, pulmonary trunk dilatation, mitral valve regurgitation and aortic root dilatation in *z*-score reported in adults than in children [8].

Contrary to the above-mentioned studies, the publication of Roman et al., in a group of 789 genetically confirmed MFS patients, no significant differences were found between children and adults with regard to external features. Nevertheless, some components of the systemic score occurred more commonly in adults, e.g., spontaneous pneumothorax, scoliosis and skin striae. Aortic complications (prophylactic aortic root replacement and aortic dissection) were rare during childhood [18].

To sum up, multiple studies on MFS have fulfilled an important role in collecting data about this rare hereditary disorder; however, this work needs to be continued. Some reports on the high positive predictive value of the combination of three phenotypic features emphasize the usefulness of physical “red flags” of MFS [13,14]. Our study highlights the need to raise awareness of these physical features of MFS. The existence of physical features, which should promptly lead to suspicion of MFS—“red flags”, should enter into clinical practice, particularly that of pediatricians, general practitioners and orthopedists. Unfortunately, these features do not have sufficient sensitivity and specificity to directly recognize MFS, but they can be utilized at different diagnostic levels, especially during screening. Coupling “red flags” with other tools, such as the Kid-SMS, may lead to enhanced diagnostics on many levels, which increases the number of patients under systematic control by cardiologists and pediatric cardiologists and decreases the number of patients waiting for detailed diagnostics in MFS centers of excellence, where patients with a high risk of complications would have priority over other patients. This may protect many patients from potentially fatal cardiovascular complications and also at the same time protect some individuals from hasty MFS suspicion and the associated psychological burden.

## 5. Conclusions

Marfan syndrome diagnosis is difficult and requires multidisciplinary assessment and not infrequently genetic testing. Still, there are many patients who were diagnosed too late, after acute aortic syndrome had occurred. On the other hand, according to our own experience, there are many individuals who are only suspected of Marfan syndrome because of high stature, which may pose significant medical and psycho-social consequences. Taking these considerations seriously, we conducted a study aimed at detecting external features, which alert to the possibility of MFS diagnosis. These so-called red flags would enable general physicians, pediatricians, orthopedists, rehabilitants, ophthalmologists and others proper selection of individuals for further specific assessment targeted at Marfan syndrome diagnosis. We elucidated “red flags” in the entire population, as well as in adults and pediatric subgroups. These “red flags” irrespective of the patient’s age are: pectus carinatum, reduced elbow extension, hindfoot deformity, gothic palate, downslanting palpebral fissures, lens subluxation, myopia ≥ 3 dioptres and remarkably high stature (excessive growth in children).

## Figures and Tables

**Figure 1 ijerph-19-00772-f001:**
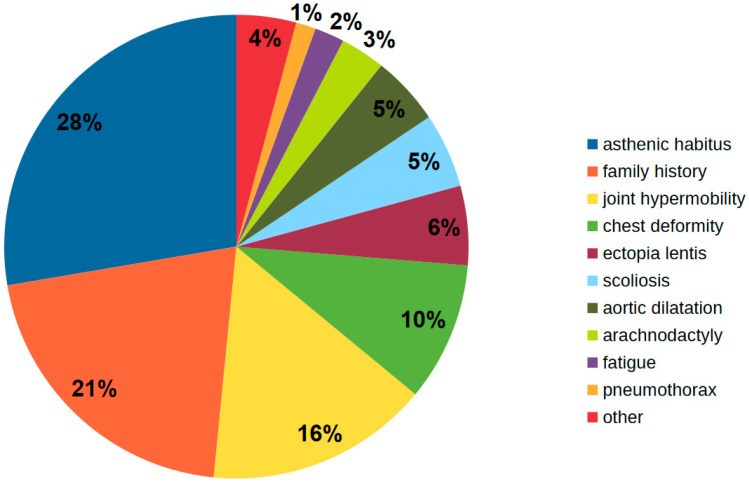
Diagram presenting reasons for Marfan syndrome suspicion.

**Figure 2 ijerph-19-00772-f002:**
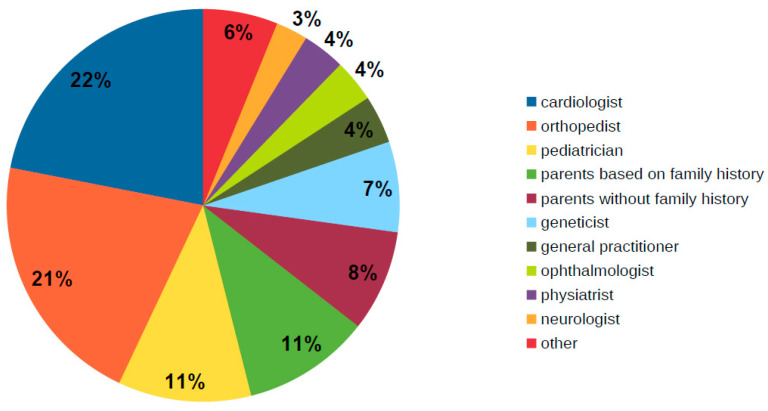
Diagram presenting specialists who were first to suspect Marfan syndrome (including parents that suspected this syndrome).

**Figure 3 ijerph-19-00772-f003:**
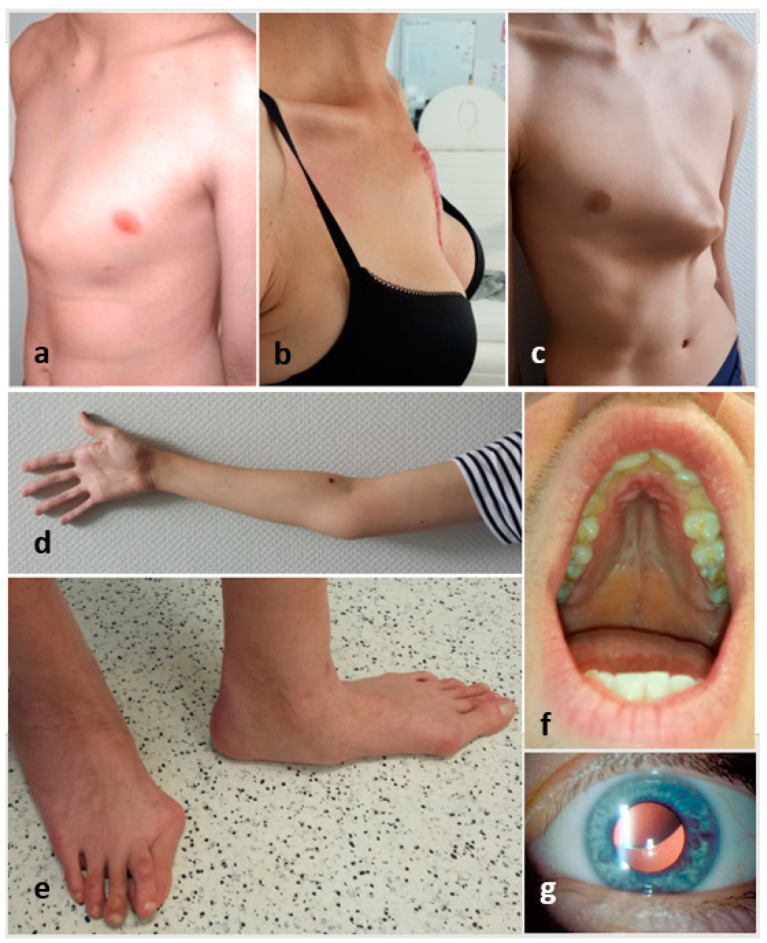
The figure shows some of the red flags described above: (**a**–**c**)—pectus carinatum, (**d**)—reduced elbow extension, (**e**)—hindfoot deformity, (**f**)—gothic palate and (**g**)—lens subluxation.

**Table 1 ijerph-19-00772-t001:** Assessment methods of some of the parameters of the physical examination.

Thumb sign	Positive when after clenching one’s fist, the distal phalanx of the adducted thumb is beyond the ulnar border of the palm (in order to reach maximal effect, assistance by the examiner is acceptable).
Wrist sign	Positive when, during the wrapping of the contralateral wrist, the thumb covers the whole nail of the fifth finger.
US/LS (upper segment to lower segment ratio)	The lower segment is measured in the standing position with the patient leaning against a wall, from the top of the symphysis pubis to the floor, whereas the upper segment is the difference between the patients’ height and the lower segment. It is considered reduced if it is <0.85.
ASHR (arm span to height ratio)	It is considered increased if it is >1.05.
Reduced elbow extension	Defined as an angle between the forearm and arm lesser than 170 (degrees).
Dolichocephaly	Defined as the ratio between the width and length of the skull, with a value ranging between 0.6–0.76.
Stretch marks	It is a clinically important sign when there is no connection between their presence and major weight fluctuations (or pregnancy) and also when they are located at an unusual area, such as the lumbar region, upper arm, mid-back or thigh.
Hindfoot deformity	It is a combination of hindfoot valgus with forefoot abduction and the lowering of the midfoot (previously referred to as the medial rotation of the medial malleolus). It should be evaluated from the anterior and posterior view and should be distinguished from the more common “flat foot” without significant hindfoot valgus.
Excessive growth (in children) remarkably high stature (in adults)	It is height equal to and above the 97th percentile in children, whilst it is equal to and above 176 cm for women and equal to and above 190 cm for men in the adult population.
Deficiency in weight	It is clinically important when BMI is under 18.5 kg/m^2^ in the adult population whilst in the child population BMI equal and under the 5th percentile.
Moderate or severe scoliosis	Scoliosis where the Cobb angle is more than 20 degrees.
Moderate or high myopia	It is myopia equal or higher than 3 dioptres

**Table 2 ijerph-19-00772-t002:** Comparative analysis of the most frequently reported symptoms, physical examination abnormalities and other more frequently noted abnormalities between patients with MFS and marfanoid habitus.

Feature	Child and Adult Population (*n* = 277)			Child Population (*n* = 178)			Adult Population (*n* = 99)		
	Marfan Syndrome (*n* = 109)	Marfanoid Habitus (*n* = 168)	*p*	Marfan Syndrome (*n* = 46)	Marfanoid Habitus (*n* = 132)	*p*	Marfan Syndrome (*n* = 63)	Marfanoid Habitus (*n* = 36)	*p*
deficiency in weight	27 (24.8)	42 (25.0)	0.959	13 (28.3)	36(27.3)	0.791	14 (22.2)	6 (16.7)	0.517
excessive growth/remarkably high stature	54 (49.5)	46 (27.4)	<0.001	19 (41.3)	33 (25.0)	0.018	35 (55.6)	13 (36.1)	0.068
ASHR >1.05	37 (35.2)	18 (10.9)	<0.001	19 (43.2)	13 (10.0)	<0.001	18 (29.5)	5 (14.3)	0.093
USLS < 0.85	95 (87.2)	155 (92.3)	0.162	40 (87.0)	122 (92.4)	0.367	55 (87.3)	33 (91.7)	0.506
joint laxity	69 (63.3)	106 (63.1)	0.972	31 (67.4)	92 (69.7)	0.771	38 (60.3)	14 (38.9)	0.040
wrist sign	57 (52.3)	68 (40.5)	0.053	25 (54.3)	56 (42.4)	0.162	32 (50.8)	12 (33.3)	0.093
thumb sign	75 (68.8)	97 (57.7)	0.064	34 (73.9)	85 (64.4)	0.238	41 (65.1)	12 (33.3)	0.002
scoliosis	86 (78.9)	119 (70.8)	0.135	35 (76.1)	92 (69.7)	0.409	51 (81.0)	27 (75.0)	0.486
moderate or severe scoliosis	15 (13.8)	15 (8.9)	0.206	3 (6.5)	12 (9.1)	0.763	12 (19.0)	3 (8.3)	0.153
pectus excavatum	27 (24.8)	42 (25.0)	0.966	16 (34.8)	38 (28.8)	0.446	11 (17.5)	4 (11.1)	0.397
pectus carinatum	44 (40.4)	22 (13.1)	<0.001	12 (26.1)	17 (12.9)	0.037	32 (50.8)	5 (13.9)	<0.001
reduced elbow extension	13 (11.9)	4 (2.4)	0.001	6 (13.0)	4 (3.0)	0.020	7 (11.1)	0 (0.0)	0.046
flat feet	64 (58.7)	91 (54.2)	0.456	31 (67.4)	76 (57.6)	0.242	33 (52.4)	15 (41.7)	0.305
hindfoot deformity	42 (38.5)	36 (21.4)	0.002	25 (54.3)	35 (26.5)	0.001	17 (27.0)	1 (2.8)	0.003
stretch marks	55 (50.5)	42 (25.0)	<0.001	16 (34.8)	32 (24.2)	0.165	39 (61.9)	10 (27.8)	0.001
dolichocephaly	45 (41.3)	68 (40.5)	0.894	26 (56.5)	59 (44.7)	0.167	19 (30.2)	9 (25.0)	0.584
dental crowding	54 (49.5)	53 (31.5)	0.003	24 (52.2)	42 (31.8)	0.014	30 (47.6)	11 (30.6)	0.097
gothic palate	71 (65.1)	43 (25.6)	<0.001	30 (65.2)	33 (25.0)	<0.001	41 (65.1)	10 (27.8)	<0.001
downslanting palpebral fissures	52 (47.7)	42 (25.0)	<0.001	24 (52.2)	35 (26.5)	0.001	28 (44.4)	7 (19.4)	0.012
retrognathia	33 (30.3)	40 (23.8)	0.233	16 (34.8)	31 (23.5)	0.134	17 (27.0)	9 (25.0)	0.829
micrognation	55 (50.5)	64 (38.1)	0.042	24 (52.2)	44 (33.3)	0.024	31 (49.2)	20 (55.6)	0.543
enophthalmia	71 (65.1)	95 (56.5)	0.154	28 (60.9)	78 (59.1)	0.832	43 (68.3)	17 (47.2)	0.039
malar hypoplasia	53 (48.6)	75 (44.6)	0.516	20 (43.5)	59 (44.7)	0.886	33 (52.4)	16 (44.4)	0.447
age at the time of first suspicion of MFS	12.9 ± 12.7	14.4 ± 7.9	0.307	5.9 ± 4.1	12.0 ± 4.1	<0.001	17.8 ± 14.3	23.6 ± 11.6	0.035
birth weight	3514.3 ± 541.0	3458.7 ± 575.8	0.575	3556.9 ± 459.8	3477.9 ± 558.2	0.456	-	-	-
birth length	55.5 ± 3.4	56.1 ± 3.3	0.405	55.9 ± 2.9	56.0 ± 3.3	0.925	-	-	-
joint pain	57 (52.3)	68 (40.5)	0.053	24 (52.2)	47 (35.6)	0.048	33 (52.4)	21 (58.3)	0.567
frequent headaches	44 (40.4)	53 (31.5)	0.133	13 (28.3)	32 (24.2)	0.589	31 (49.2)	21 (58.3)	0.382
dizziness	34 (31.2)	24 (14.3)	0.001	6 (13.0)	13 (9.8)	0.582	28 (44.4)	11 (30.6)	0.174
syncope	30 (27.5)	31 (18.5)	0.075	9 (19.6)	23 (17.4)	0.745	21 (33.3)	8 (22.2)	0.243
chest pain	37 (33.9)	42 (25.0)	0.107	8 (17.4)	27 (20.5)	0.653	29 (46.0)	15 (41.7)	0.674
palpitations	27 (24.8)	24 (14.3)	0.028	5 (10.9)	14 (10.6)	1.000	22 (34.9)	10 (27.8)	0.465
coordination disorders	38 (34.9)	59 (35.1)	1.000	18 (39.1)	49 (37.1)	0.809	20 (31.7)	10 (27.8)	0.679
hernias	26 (23.9)	16 (9.5)	0.001	9 (19.6)	12 (9.1)	0.058	17 (27.0)	4 (11.1)	0.063
idiopathic pulmonary oedema	6 (5.5)	3 (1.8)	0.161	2 (4.3)	2 (1.5)	0.275	4 (6.3)	1 (2.8)	0.650
multiple injuries	15 (13.8)	19 (11.3)	0.543	4 (8.7)	11 (8.3)	1.000	11 (17.5)	8 (22.2)	0.563
effort tolerance worse than that of peers	54 (49.5)	57 (33.9)	0.010	19 (41.3)	37 (28.0)	0.095	35 (55.6)	20 (55.6)	1.000
lens subluxation	45 (41.3)	2 (1.2)	<0.001	18 (39.1)	1 (0.8)	<0.001	27 (42.9)	1 (2.8)	<0.001
myopia ≥ 3 dioptres	43 (39.4)	21 (12.5)	<0.001	14 (30.4)	13 (9.8)	0.001	29 (46.0)	8 (22.2)	0.018

**Table 3 ijerph-19-00772-t003:** Multivariate logistic analysis in order to establish independent predictors for the diagnosis of MFS.

Feature	Child and Adult Population (*n* = 277)			Child Population (*n* = 178)			Adult Population (*n* = 99)		
	*p*	HR	95%CI	*p*	HR	95%CI	*p*	HR	95%CI
ASHR > 1.05	0.035	3.48	1.09–11.11	0.003	11.48	2.29–57.48	0.810	0.73	0.06–9.13
thumb sign	0.643	1.32	0.41–4.20	0.888	1.13	0.21–6.03	0.269	3.78	0.36–40.07
wrist sign	0.939	0.96	0.32–2.84	0.875	0.88	0.18–4.36	0.622	0.56	0.05–5.74
pectus carinatum	0.048	3.00	1.01–8.92	0.808	0.80	0.13–4.94	0.131	4.75	0.63–35.79
scoliosis	0.557	1.40	0.45–4.35	-	-	-	-	-	-
assymetry of the chest	0.410	1.58	0.53–4.65	0.119	4.33	0.69–27.38	-	-	-
joint laxity	-	-	-	-	-	-	0.179	5.82	0.45–76.04
reduced elbow extension	0.816	1.27	0.17–9.60	0.396	3.02	0.24–38.59	0.999	168.49	0.76–9605.20
enophthalmia	0.381	1.53	0.59–3.93	-	-	-	0.040	9.60	1.11–82.89
micrognation	0.336	0.62	0.23–1.64	0.545	0.58	1.00–3.39	-	-	-
retrognathia	-	-	-	0.919	0.91	0.15–5.63	-	-	-
downslanting palpebral fissures	0.167	1.94	0.76–4.98	0.152	2.71	0.69–10.58	0.807	0.77	1.00–6.23
gothic palate	0.221	1.79	0.71–4.53	0.534	1.59	0.37–6.78	0.908	1.13	0.14–9.42
dental crowding	0.403	1.52	0.57–4.06	0.228	2.43	0.57–10.26	0.361	2.60	0.34–20.18
stretch marks	0.159	2.04	0.76–5.50	-	-	-	0.114	5.11	0.68–38.53
hindfoot deformity	0.005	4.54	1.57–13.20	0.017	6.02	1.39–26.15	0.119	10.61	0.55–206.61
myopia ≥ 3 D	0.224	2.00	0.65–6.15	0.732	0.67	0.07–6.59	0.804	1.31	0.16–10.82
lens subluxation	<0.001	67.12	6.51–691.63	0.001	78.91	5.51–1130.44	0.998	364.67	0.78–3698.90
excessive growth/remarkably high stature	0.001	5.46	1.99–14.93	0.076	3.87	0.87–17.25	0.044	14.54	1.07–197.83
hernias	0.004	6.45	1.83–22.67	0.044	7.25	0.87–17.25	0.574	2.16	0.15–31.86
idiopathic pulmonary oedema	0.758	1.43	0.15–14.10	-	-	-	-	-	-

**Table 4 ijerph-19-00772-t004:** Features that turned out to be useful for differentiating between Marfan syndrome and marfanoid habitus in children and adults should be seen as red flags in the context of Marfan syndrome suspicion.

Children	Adults
pectus carinatum	
reduced elbow extension	
hindfoot deformity	
gothic palate	
downslanting palpebral fissures	
lens subluxation	
myopia ≥ 3 dioptres	
excessive growth/remarkably high stature	
ASHR > 1.05	thumb sign
dental crowding	joint laxity
micrognation	stretch marks
hernias	enophthalmia

## Data Availability

Not applicable.

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
