# Peer review of "How to Distinguish Marfan Syndrome from Marfanoid Habitus in a Physical Examination—Comparison of External Features in Patients with Marfan Syndrome and Marfanoid Habitus"

_ijerph, 2022, doi:10.3390/ijerph19020772_

Round 1

Reviewer 1 Report

Authors assessed the external features of Marfan Syndrome in child and adult population and tried to present the red flags. Authors tried well to find and differentiate some features between children and adults. I am actually quite surprised that they only used modified Ghent criteria in today's age for analysis and discriminating between Marfan syndrome and marfanoid habitus, but didn't present molecular confirmation and its correlation. Molecular testing can confirm upto 98% of individuals suspected to have True Marfan syndrome. Revised Ghent criteria actually is a diagnostic and probabilistic tool, but ultimately molecular testing can confirm the diagnosis. This paper adds very limited information to what is existing and I would suggest a correlation with genetic testing shall be done with individual features to make it a a stronger manuscript. 

Please write to expand the abbreviation's full form where used first time for MASS phenotype.

Author Response

Dear Reviewer,

Thank you very much for your careful review and comments. We appreciate your evaluation for the improvement of our manuscript. We have made the following revisions in response to your comments:

1)    We fully agree with your suggestion about molecular confirmation of Marfan syndrome. Unfortunately, it (NGS) is available almost exclusively commercially and requires many months of waiting for the results. For this reasons, in Poland genetic testing is performed only in patients for whom the Ghent criteria do not give a conclusive answer. Therefore, since the retrospective nature of the present study we had to rely mainly on Ghent criteria as differentiating tool. We plan to extend our research by adding valuable molecular diagnostics in the future.

2) Thank you for this suggestion. We have made corrections in accordance to it on page 5.:

MASS phenotype is a familial connective tissue disorder similar to Marfan syndrome. MASS is an acronym for the features that include: M – mitral valve prolapse, A – aortic root dilation, S – skin striae, S – skeletal features).

Reviewer 2 Report

This manuscript is a very interesting study by Wozniak-Mielczarek et al, where the authors attempted to distinguish between the MFS and Marfanoid Habitus by physical examination. This study could have a very important clinical angle that can identify patients with MFS predisposition. The study is done thoroughly following that goal. I think this study will benefit the scientific community. Hence, this could be a nice addition to MDPI-IJERPH. However, I do think that the outcome of the study is not very clear and needs further clarifications. Keeping that in mind, I have the following comments. If the other reviews are favorable, I would like the authors to explain them in the discussion section, before this can be published.

(1) For the adult population that the authors have studied, they didn’t divide them in female and male category. Could this phenomenon be affected by gender of the adults? How about the age of the adults as well?

(2) The authors should describe and comment about the outcome and the future impacts of the study with further details with the prospective future benefits. Right now, I agree that they have few things about this in different parts of the text, but that is rather vague. What do they want to conclude? Are the final observations that they made align with the initial goal of their study? This needs to be clarified in details.  

Author Response

Dear Reviewer,

Thank you very much for your careful review and comments. We appreciate your evaluation for the improvement of our manuscript. We have made the following revisions in response to your comments:

  • For the adult population that the authors have studied, they didn’t divide them in female and male category. Could this phenomenon be affected by gender of the adults? How about the age of the adults as well?

Thank you for your relevant remark. We agree with your suggestion, therefore we have calculated and added to the manuscript comparative analysis between women and men in the entire adult population, in the population of adults with Marfan syndrome and in the population of adults with marfanoid habitus. We also conducted a comparative analysis between patients with Marfan syndrome and marfanoid habitus separately in the group of women and men. We added this information to the manuscript and added the prepared tables to supplementary material (Tables S4-S8).

(2) The authors should describe and comment about the outcome and the future impacts of the study with further details with the prospective future benefits. Right now, I agree that they have few things about this in different parts of the text, but that is rather vague. What do they want to conclude? Are the final observations that they made align with the initial goal of their study? This needs to be clarified in details.

Thank you for this valuable suggestion. According to your suggestion, we added following sentences in conclusions sections:

Marfan syndrome diagnosis is difficult and requires multidisciplinary assessment and not infrequently genetic testing. Still, there are many patients, who are diagnosed too late, after acute aortic syndrome had occurred. On the other hand, according to our own experience, there are many individuals who only due to high stature are suspected of Marfan syndrome, which may pose significant medical and psycho-social consequences. Taking these considerations seriously, we conducted study aimed at detecting external features, which alert to the possibility of MFS diagnosis. These, so called, red flags, would enable general physicians, pediatricians, orthopedists, rehabilitants, ophthalmologists and others proper selection of individuals for further specific assessment targeted at Marfan syndrome diagnosis. We elucidated “red flags” in the entire population, as well as in adults and pediatric subgroups. These “red flags” irrespective of patients age are: pectus carinatum, reduced elbow extension, hindfoot deformity, gothic palate, downslanting palpebral fissures, lens subluxation, myopia ≥ 3 D, remarkably high stature (excessive growth in children).

Round 2

Reviewer 1 Report

Thanks for the changes. I think, I would still like to know that among whatever patients you did molecular and came positive, how many were MFS and how many were Marfanoid like.

Author Response

Dear Reviewer,

Once again, thank you very much for your careful review and comment. Among 306 patients we have examined, genetic testing was performed in 35: in 11 the diagnosis of Marfan syndrome was confirmed, in 7 the diagnosis of Loeys-Dietz syndrome was confirmed, in 5 the diagnosis of Ehlers-Danlos was confirmed and in 12 patients no pathogenic or potentially pathogenic mutation was found (these patients were diagnosed for mutations in the FBN1 gene).
We have added this information to the manuscript in the Patient Characteristics section.